

# Differential expression of some termite neuropeptides and insulin/IGF-related hormones and their plausible functions in growth, reproduction and caste determination

Jan A. Veenstra

INCIA UMR 5287 CNRS, Université de Bordeaux, Pessac, France

## ABSTRACT

**Background**. Insulin-like growth factor (IGF) and other insulin-like peptides (ilps) are important hormones regulating growth and development in animals. Whereas most animals have a single female and male adult phenotype, in some insect species the same genome may lead to different final forms. Perhaps the best known example is the honeybee where females can either develop into queens or workers. More extreme forms of such polyphenism occur in termites, where queens, kings, workers and soldiers coexist. Both juvenile hormone and insulin-like peptides are known to regulate growth and reproduction as well as polyphenism. In termites the role of juvenile hormone in reproduction and the induction of the soldier caste is well known, but the role of IGF and other ilps in these processes remains largely unknown. Here the various termite ilps are identified and hypotheses regarding their functions suggested.

**Methods**. Genome assemblies and transcriptome short read archives (SRAs) were used to identify insulin-like peptides and neuropeptides in termites and to determine their expression in different species, tissues and castes.

**Results and Discussion**. Termites have seven different ilps, *i.e.* gonadulin, IGF and an ortholog of *Drosophila* insulin-like peptide 7 (dilp7), which are commonly present in insects, and four smaller peptides, that have collectively been called short IGF-related peptides (sirps) and individually atirpin, birpin, cirpin and brovirpin. Gonadulin is lost from the higher termites which have however amplified the brovirpin gene, of which they often have two or three paralogs. Based on differential expression of these genes it seems likely that IGF is a growth hormone and atirpin an autocrine tissue factor that is released when a tissue faces metabolic stress. Birpin seems to be responsible for growth and in the absence of juvenile hormone this may lead to reproductive adults or, when juvenile hormone is present, to soldiers. Brovirpin is expressed both by the brain and the ovary and likely stimulates vitellogenesis, while the function of cirpin is less clear.

Corresponding author
Jan A. Veenstra, jan-adrianus.veenstra@u-bordeaux.fr

## INTRODUCTION

Termites are fascinating insects. They have long-living reproductives that are called queens and kings and their descendants constitute large colonies. Many of their descendants are sterile, with some of them having exclusive defensive functions, the so-called soldiers and others, the workers, take care of food collection, which in some species includes cultivating specific fungi. The royal pair is in charge of most of the reproduction of the colony. In some species a termite queen can lay more than 10,000 eggs per day, a quantity that is possible because of her specialized physiology and morphology. Termite queens have very large abdomens, a phenomenon known as physogastry (*Grassé, 1982*; *Wilson, 1971*). Why and how the same genome produces such different phenotypes is a fascinating question.

Termites evolved from more basal cockroaches and their evolution can be illustrated by the branches of a phylogenetic tree (Fig. 1). *Periplaneta americana* is found at the root of the tree and on the first branch is the wood eating *Cryptocercus* cockroach where adults care for their young. Moving further up, the first termite is *Mastotermes darwiniensis*, a species that lays eggs in a structure resembling a cockroach ootheca. *Zootermopsis nevadensis* and *Hodotermopis sjostedti* are found a little higher but are still relatively primitive termites with colony sizes of up to 4,000 members. *Cryptotermes secundus* is more developed and, although the average size of its colonies is around 300-400, individuals colonies can also reach several thousand members (*Korb & Lenz, 2004*). *Reticulitermes speratus* and *Coptotermes formosanus* are still higher up on the tree, but neither is a member of the most advanced termite family, the Termitidae, to which *Macrotermes natalensis* belongs. It is *Macrotermes* queens that can lay such large numbers of eggs that they can create colonies with hundreds of thousands of members.

The couple that starts a colony is referred to as the royal pair, a queen and a king that are also known as the primary reproductives. Their descendants can develop into workers or soldiers that are sterile, or become members of a reproductive caste. Soldiers have a defensive function that is facilitated by their distinctively enlarged mandibles, which are useful for defense but make eating difficult and, consequently, soldiers are fed by workers. Some higher termites may have two classes of soldiers and/or workers, which are usually referred to as minor and major (*Wilson, 1971*; *Grassé, 1982*).

Neither workers nor soldiers have wings or wing buds. Nymphs on the other hand are larvae that may develop into winged males and females, the new generation of primary reproductives, which after their final molt leave the nest to start a new colony. Many species have secondary reproductives, also known as neotenic reproductives, that can develop from workers or soldiers under specific conditions. This transformation of larvae into different termite castes varies from one species to another. Whereas in the lower termites the final destination of a larvae is determined late in its development and can be reversed, thus revealing great flexibility, in the higher termites this is determined earlier during development, and lacks such flexibility (*Wilson, 1971*; *Grassé, 1982*).

Termites are not the only insects with polyphenism, it also exists in other insects, such as ants, honey bees, aphids, plant hoppers and crickets. In many of the latter species signaling by insulin-related peptides and juvenile hormone play essential roles in deciding whether

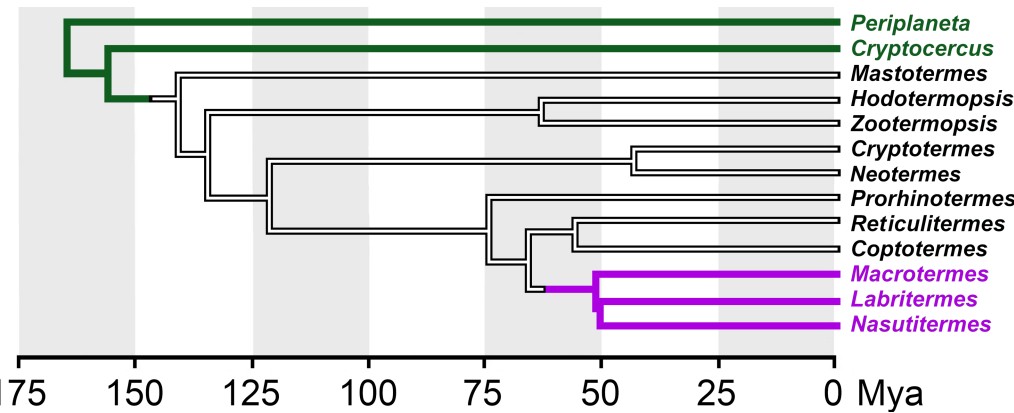

**Figure 1** Simplified phylogenetic tree showing the relationships among termite genera and other cockroaches. Tree illustrating relations between the various termites analyzed here and closely related cockroaches. This tree is based on data from *Bucek et al. (2019)* and *Evangelista et al. (2019)*. Green, cockroaches, magenta higher termites (termitidae). Mya, millions years ago.

the individual will develop into one or another form (*e.g.*, *Goewie & Beetsma, 1976*; *Wheeler, Buck & Evans, 2006*; *Xu et al., 2015*; *Zera, 2016*; *Chandra et al., 2018*; *Cuti et al., 2021*; *Yan et al., 2022*). In termites it is known that juvenile hormone will induce soldiers, while the insulin/IGF signaling pathway is suspected to play an important role as well (*e.g.*, *Hattori et al., 2013*; *Kremer, Korb & Bornberg-Bauer, 2018*).

Expression of these hormones is partly controlled by the diet (*Corona, Libbrecht & Wheeler, 2016*). Production of IGF and juvenile hormone is increased when honeybee larvae are fed royal jelly, produced by specialized exocrine glands of nurse bees. Food is also extensively shared by termites using both stomodaeal and proctodaeal trophollaxis. Proctodaeal trophollaxis is necessary for the transfer of essential microbial mutualists, while in stomodaeal trophollaxis both partially digested wood particles and salivary proteins are transferred. In all termites reproductives are fed exclusively with saliva produced by the workers and in higher termites this is also the case for larvae and nymphs (*Grassé, 1982*).

The availability of a large number of publicly available termite short read archives (SRAs) for termite transcriptomes and five termite genome assemblies (*Harrison et al., 2018*; *Korb et al., 2015*; *Terrapon et al., 2014*; *Itakura et al., 2020*; *Shigenobu et al., 2022*) makes it possible to explore both the existence and expression of insulin-related hormones in termites. Here, I have used these genome assemblies to identify the various termite insulin-related peptides and their putative receptors and the transcriptome SRAs to get a preliminary idea of their expression, which might provide indications as to their functional relevance. In order to understand the functions of the various insulin related peptides some proteins that might provide clues as to the endocrinological and physiological status of the insects analyzed were also studied.

## MATERIAL AND METHODS

### Transcript identification

Genome assemblies were downloaded from NCBI (https://www.ncbi.nlm.nih.gov/genome/?term=blattodea) and GigaDB (http://gigadb.org/dataset/100057). The methodology of the identification of coding sequences for neuropeptide precursors, insulin-related hormones and other proteins has been described in detail elsewhere (*Veenstra, 2020*; *Veenstra, 2021*). It consists of using a combination of the BLAST+ program (https://hpc.nih.gov/apps/Blast.html), the sratoolkit (https://hpc.nih.gov/apps/sratoolkit.html), Artemis (*Rutherford et al., 2000*) and Trinity (*Grabherr et al., 2011*) on available genome assemblies and SRAs to construct complete cDNAs. As a query, I used sequences homologous to *Z. nevadensis* neuropeptides (*Veenstra, 2014*), as well as the *Z. nevadensis* orthologs of several neuropeptide precursors that were more recently identified (*Liessem et al., 2018*; *Xie et al., 2020*; *Veenstra, 2020*; *Zeng et al., 2021*). A recently described putative neuropeptide precursor from *Periplaneta americana* (*Zeng et al., 2021*) has been baptized fliktin in ants (*Habenstein et al., 2021*) and this name was used here also. Whereas it is possible that a neuropeptide has mutated so much that it escapes detection in a BLAST search, is becomes statistically impossible for the transmembrane regions of its G-protein coupled receptor (GPCR) to leave not a single read in an SRA that covers a genomic sequence many-fold (*Veenstra, 2019*). This rationale was applied to demonstrate that the gonadulin signaling pathway is genuinely absent from several termite species and also to show that ACP was lost in both *Reticulitermes speratus* and *Macrotermes natalensis*. In both cases the protein sequence of the predicted transmembrane region of the orthologous *Zootermopsis* GPCR (*Veenstra, 2014*) was cut in pieces corresponding to the coding exons as determined in the *Zootermopsis* genome. These protein sequences were then used as query in the vdb_tblastn command to search for reads in the respective genome SRAs.

The following proteins were also analyzed: vitellogenin, vitellogenin receptor, hexamerins and neural lazarillo (*Hagedorn & Kunkel, 1979*; *Tufail & Takeda, 2009*; *Haunerland, 1996*; *Yaguchi et al., 2018*). Vitellogenins are the major yolk proteins and their synthesis may indicate a reproductive female, while the vitellogenin receptor is exclusively located in the ovary. Hexamerins are storage proteins that serve as a depot for amino acids, which are recycled when these proteins are degraded. There are two major types, tyr-hexamerin which is rich in tyrosine and phenylalanine residues, and met-hexamerin. Synthesis of tyr-hexamerin increases before each molt and after molting its degradation provides amino acids for cuticle proteins that are rich in tyrosine. Met-hexamerin is enriched in methionine residues and its synthesis is usually increased before the adult molt; its amino acids contribute to the production of novel adult structures, such as the sexual organs. Neural lazarillo, a lipocalin, was identified as a protein that was suggested to be crucial for soldier determination (*Yaguchi et al., 2018*).

All coding sequences and their conceptual translations are presented in Spreadsheet S1.

### SRAs

The complete list of SRAs used in the analysis can be found in the Supplementary Data. All SRAs were downloaded from NCBI (https://www.ncbi.nlm.nih.gov/sra/). Unfortunately,

a large set of *R. speratus* SRAs could not be used for expression analysis as they appear to be incorrectly labeled. For example DRR266556, DRR266558 and DRR266559 contain very large numbers of brovirpin reads but lack reads for neuroparsin, AKH, SIFamide and SMYamide, genes that are almost exclusively expressed in the brain or corpus cardiacum. These are therefore likely all are queen body transcriptomes, yet their descriptions at the database state something else. SRR3178362 is likely contaminated, as it contains spots for a neuroparsin that is absent from all other genome or transcriptome *Macrotermes* SRAs. It is somewhat surprising that there are no ovary specific transcriptome SRAs from well studied species such *C. secundus, Z. nevadensis* or any of the *Macrotermes* or *Reticulitermes* species and there also relatively few soldier specific SRAs. While there is a single salivary gland SRA from *Coptotermes formosanus*, comparative data from a higher termite such as a *Macrotermes* species is unfortunately lacking.

## Quantification and statistics

In principle, it should be possible to use SRAs to look for differences in expression between different castes and to identify potential hormones, proteins and/or neuropeptides of interest. This requires quantitative analysis of gene expression. Most of the transcriptome data analyzed here was obtained by using polyA selected material. This creates a problem in that random breakage of mRNA will lead to an enrichment of reads coding for the C-terminal parts of proteins, whereas those coding for the N-terminal of proteins will be rarer. For large proteins, as well as those that have mRNAs with large untranslated $3'$-sequences, the total number of reads will thus be an underestimate of the actual mRNA levels. I therefore did not translate expression into fragments or reads per kilobase of million mapped reads, but used the numbers of spots in each SRA. The number of gene specific spots in each SRA was obtained by using the blastn_vdb command using the coding sequence of its transcript as query and options -outfmt 6 sseqid and -max_target_seqs 10000000. Duplicates were removed from result files using the sort -u linux command and spots counted. This procedure works well for little expressed genes, such as insulin-like homones, neuropeptides and their receptors, but does not work with strongly expressed proteins like vitellogenin in queens. Read counts for these proteins were obtained by counting a sample from the SRA. This was done using the fastq-dump command from the sratoolkit to extract the fasta sequences and then building blastdb from the first 1,000,000 lines of the fasta file and using the blastn command from BLAST+ following the same procedure as outlined above but using BLAST+.

It is not obvious to compare tissues or body parts from different castes. First, it is not clear that in animals with a different morphology, *e.g.*, workers and queens, or workers and soldiers, the percentages of the fat body and muscle will be the same. Indeed it is well known that brain sizes are markedly different between the castes (*O'Donnell, Bulova & Barrett, 2021*). Furthermore, based on the presence of reads for neuropetides typically expressed by the central nervous system, some SRAs that are labeled as fat body appear to contain in addition the ventral cord and perhaps even the brain, while others do not. Castes do not only differ in size and morphology, but also in their diets. While the analysis was in progress, it was realized that another potential problem is nutritional stress. Whereas

the royal pair is well fed, at least the workers and perhaps soldiers are comparatively malnourished. When analyzing queens from large termite colonies, the time between the moment that the queen is no longer fed by its workers and when the animal is dissected may well be long enough to induce starvation. A queen that is virtually a machine producing eggs needs to be continuously fed and once feeding stops, starvation may develop rapidly. Some of such queen samples show increased production of AKH and atirpin, which may both function as stress hormones, as explained below.

Comparisons between the relative numbers of spots found for a particular gene in one SRA *versus* those of the same gene in another SRA seems straightforward, however the data suggests this may be naive and likely yields misleading results. Thus, the relative spot numbers for neuropeptide and insulin-like peptide (ilp) spots in minor and major presoldiers of *M. barneyi* are on average 1.79 times higher in minor than in major presoldiers; a difference that is consistent for the three replicates of these samples. The same difference between larvae predestined to become minor and major workers is almost as high at 1.59 times. It makes no sense to assume that larvae destined to stay smaller—minor *versus* major in both cases—would need larger quantities of neuropeptides, indeed such differences are not found for five household genes (Spreadsheet S2, *M. barneyi*-1). It is a more reasonable hypothesis to assume that larvae that are destined to become bigger will need more structural proteins and, consequently, the fraction of the neuropeptides spots in SRAs from such larvae will be a smaller fraction of the total. Similar differences can be expected between workers and reproductive females that produce large quantities of vitellogenin.

As an alternative to using the number of spots for a specific gene divided by the total number of spots in an SRA I have divided the number of reads for a specific neuropeptide by the total number of all neuropeptide reads from that particular SRA. This yielded more consistent results (Spreadsheet S2, *M. barneyi*-2). However, the latter approach is only valid if a neuropeptide is expressed exclusively within the nervous system. Some insulin-like peptides—IGF, atirpin and brovirpin—and AKH are produced in peripheral tissues. In those comparisons where one or more of these neuropeptides yielded high numbers or RNAseq reads (there are often samples with high spot numbers for AKH and atirpin), their numbers were excluded from the total number of neuropeptide reads. This illustrates the inherent difficulties in making quantitative comparisons.

Even when quantitative comparisons appear to be valid, reliable statistical analysis of such data is not easy. One SRA essentially yields $n = 1$. As a consequence of the expense involved in NGS sequencing numbers are often small and not all SRAs necessarily correspond to independent samples; some SRAs have identical content (*e.g.*, ERR2615953/SRR5457739 and ERR2615954/SRR5457740), while others are so similar in the relative numbers of reads for genes that they must have originated from the same initial sample (*e.g.*, ERR2615943/SRR5457745, ERR2615944/SRR5457744, ERR2615945/SRR5457743 and ERR2615946/SRR5457742). In many cases the number of observations are often small, $n1 = n2 = 3$ or 5. Thus although in many cases differences are quantitated, it is virtually impossible to apply a valid statistical test. Wilcoxon's rank sum test was used to get an indication as to how significant some of the differences in neuropeptide expression between
6 month old male and female reproductives in *R. speratus* could be as there are 15 replicates in that case.

## Sequence similarity and phylogenetic trees

Both phylogenetic and sequence similarity trees use Clustal omega (*Sievers et al., 2011*) to produce alignments. Fasttree (*Price, Dehal & Arkin, 2010*), using the ./FastTreeDbl command with the -spr 4, -mlacc 2 and -slownni options, was used to construct trees and estimate probabilities.

# RESULTS AND DISCUSSION

## Identification of insulin-like ligands

The publicly available genomes of two basal cockroach and five termite species—*Blattella germanica*, *Periplaneta americana*, *Zootermopsis nevadensis*, *Coptotermes formosanus*, *Cryptotermes secundus*, *Reticulitermes speratus* and *Macrotermes natalensis* (*Harrison et al., 2018*; *Korb et al., 2015*; *Terrapon et al., 2014*; *Itakura et al., 2020*; *Li et al., 2018*; *Shigenobu et al., 2022*)—allow the identification of the genes coding for insulin-like peptides and their putative receptors.

Termites typically have three loci in their genomes coding for insulin-related peptides. The first contains genes coding for gonadulin, IGF and a dilp7 (drosophila insulin-like peptide 7) ortholog. The last common bilaterian ancestor of the protostomes and deuterostomes must already have had such a triplet of insulin/IGF-related genes, as orthologous gene triplets are not only present in arthropods but also in ambulacrians (*Veenstra, 2021*). While both gonadulin and the dilp7 ortholog have been lost from several species, IGF seems to be commonly present in all insects, even though its structure in *Drosophila* has been greatly reduced during evolution (*Veenstra, 2020*). The primary structures of termite IGF and the dilp7 orthologs are very well conserved, but this is less so for their gonadulins (Figs. S1–S3). Most of these genes seem functional, but in *Macrotermes* and *Reticulitermes* species the presence of in-frame stop codons show that the gonadulin genes have become pseudogenes. No gonadulin gene was identified in the *Coptermes formosanus* genome. Obviously in some species the gonadulin gene is no longer functional. As in other insect species (*Veenstra, 2020*) the termite IGF genes yield two alternatively spliced transcripts that code for slightly different mature peptides. These three genes—gonadulin, IGF and dilp7 ortholog—are found next to one another in a characteristic conformation in the genomes of *Blattella germanica, Zootermopsis nevadensis, Cryptotermes secundus, Macrotermes natalensis, Coptotermes formosanus* and *Reticulitermes speratus,* as previously described for *B. germanica* and *Z. nevadensis* (*Veenstra, 2020*).

There are two other genome loci with genes coding for insulin-like peptides; those genes likely evolved from a non-local gene duplication of the IGF gene (*Veenstra, 2021*). I will call the peptides coded by these genes sirps, for short IGF-related peptides, and use the ilp acronym for the ensemble of IGF, gonadulin, dilp7 orthologs and sirps. In both cockroaches and termites the majority of the sirp genes are located next to one another in the genome, suggesting that they evolved through subsequent local gene duplications. Insect sirp sequences are notoriously variable and it is generally difficult to establish

homology between sirps from different species unless they are closely related. A sequence similarity tree of the termite sirp sequences (Fig. 2) reveals four major branches. This tree suggests the different sirp genes to be orthologous and this is confirmed by the synteny of their genes (Fig. 3). In all termite genome assemblies one sirp gene (cirpin) is located on a different chromosomal fragment than the others, but this could be due to genome assembly problems. It is clear that the amino acid sequences of the various termite sirps are well conserved and each may well have its own specific function. I have therefore baptized them as atirpin, birpin, brovirpin and cirpin (these names are derived from **a**ll-**t**issues-**irp**, **b**rain-**irp**, **br**ain-**ov**ary-**irp** and **c**erebral-**irp**, as based on their expression as described below; it is likely that these names will be useful only with termites). Atirpin and birpin are well conserved, while cirpin and brovirpin are more variable (Figs. S4–S7). The brovirp gene has been amplified in higher termites, suggesting it may be more important in those species.

As termites are evolved cockroaches, it is of interest to see how the various termite peptides relate to those of the the more basal cockroaches. Basal cockroaches typically have a larger number of sirps and when these are added to the tree, both the atirpin and birpin branches are again well defined, but the brovirpin and cirpin branches are no longer well resolved (Fig. S8). Basal cockroaches seem to have a single atirpin, often two birpins and a number of other sirps that are somewhat similar to termite cirpin and brovirpin. When comparing the genome sequences of *Zootermopsis,* the most basal termite for which a genome assembly is available, with that of *Periplaneta*, a species that is relatively closely related to termites (Fig. 1), it is obvious that several sirp genes have been lost in termites. In transcriptomes of both *Mastotermes*, the most basal termite, and *Cryptocercus,* its nearest non-termite relative, only orthologs of the four *Zootermopsis* sirps can be identified, suggesting that the loss of the other cockroach sirps occurred before termites evolved.

Although the cockroach sirps have very variable amino acid sequences, atirpin has structural characteristics that sets it apart from the other sirps and in which it is in fact more similar to IGF. The second and third cysteine residues in the A chain of both IGF and atirpin are separated by three amino acid residues and not four as in the other sirps (Fig. 4). This is part of the core sequence of ilps and can be expected to have consequences for receptor binding. Furthermore, whereas most sirps have typical KR neuroendocrine cleavage sites that separate the putative connecting peptides from the predicted A- and B-chains of these peptides, such sites are not as easily identified in the atirpin sequences while the sequence corresponding to the connecting peptide in atirpins is much better conserved, both in length and amino acid composition. These characteristics suggest that atirpin, like IGF, may be processed through the constitutive pathway.

## Identification of neuropeptides

Coding sequences for neuropeptides were deduced for some but not all species. The neuropeptide precursors are in general very similar to those previously described from *Z. nevadensis* (*Veenstra, 2014*). Since then several other insect neuropeptides have been described, these are *Carausius* NPLP, hansolin, RFLamide, iPTH and fliktin (*Liessem et al., 2018*; *Xie et al., 2020*; *Zeng et al., 2021*) which are also present in termites. ACP

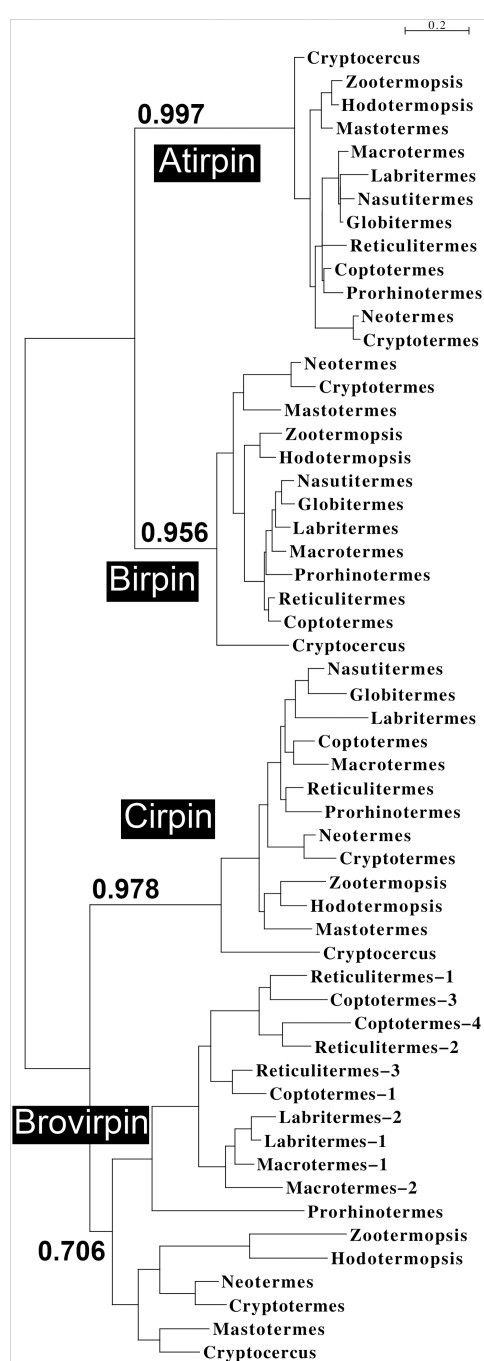

**Figure 2 Comparison of the sequences of the various termite sirps identifies four different types.** Sequence similarity tree of sirps form termites and *Cryptocercus*, the non-termite cockroach species most closely related to termites. Four different branches, corresponding to atirpin, birpin, cirpin and brovirpin are present in this tree.

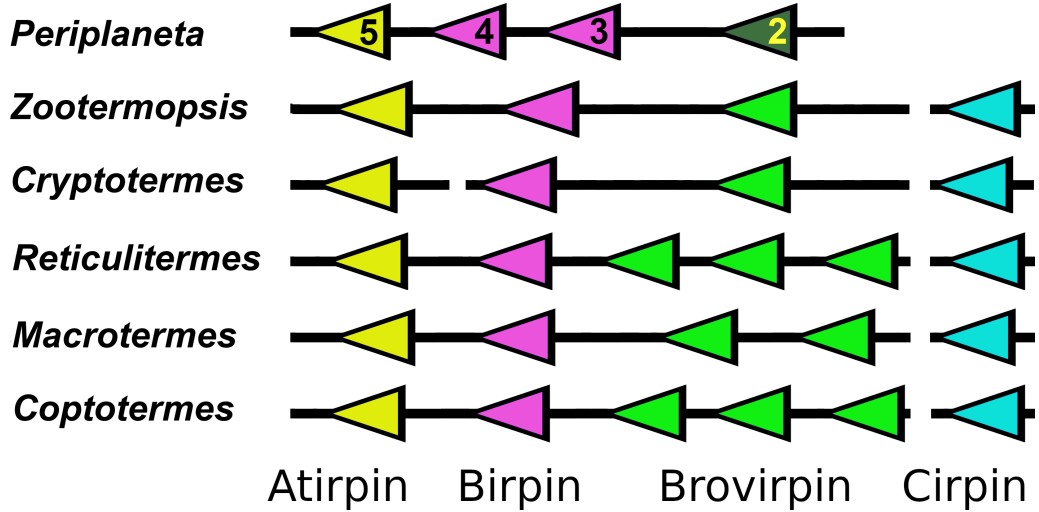

**Figure 3 Synteny of sirp genes.** Comparative genome localizations of sirp genes in five termites and *Periplaneta americana*. Note that the cirpin gene is always found on a different scaffold, as indicated by the interruption of the black lines. Arrows indicate direction of transcription of the genes. Similar colors indicate similar amino acid sequences. The numbers of the *Periplaneta* genes correspond to the insulin-like peptides previously described from this species (*Zeng et al., 2021*).

(Adipokinetic hormone/Corazonin-related peptide) and its receptor are missing from the *R. speratus, C. formosanus* and *M.natalensis* genome assemblies. Using the vdb_tblastn command with the *Z. nevadensis* ACP receptor to query the genomic SRAs from *R. speratus* and *M. natalensis* yielded reads corresponding to the AKH and corazonin receptors. Confirming that the ACP signaling system has been lost from this species. Genome SRAs for *C. formosanus* are not publicly available, but it seems likely that ACP was already lost in the last common ancestor of these termites.

It was previously shown that the *Z. nevadensis* allatostatin CC precursor contains a signal anchor rather than a signal peptide (*Veenstra, 2014*). A signal anchor was also found in the allatostatin CC precursor from the other termites, indicating that, as in *Drosophila* (*Veenstra, 2009*), in termites this neuropeptide functions as a juxtracrine, rather than a para- or endocrine. In *Periplaneta* the allatostatin CC has a normal signal peptide, but already in the *Cryptocercus* species there is a signal anchor. While the *M. natalensis* genome sequence suggests that the CNMamide gene has still two functional transcripts, both the *C. secundus* and *R. separatus* genome assemblies show only one CNMa transcript. The *C. formosanus* genome sequence shows the remnants of the second transcript but, since it has lost the convertase cleavage site at the C-terminal of the putative mature neuropeptide, it is most likely no longer functional.

### Expression and functional relevance of Gonadulin and Dilp7 orthologs

Significant gonadulin expression is found in neotenic males and females of *Hodotermopsis* and *Zootermopsis* (Spreadsheet S2, *Z. nevadensis*-1), both primary and secondary reproductive females as well as alate females, where expression is somewhat lower. Males

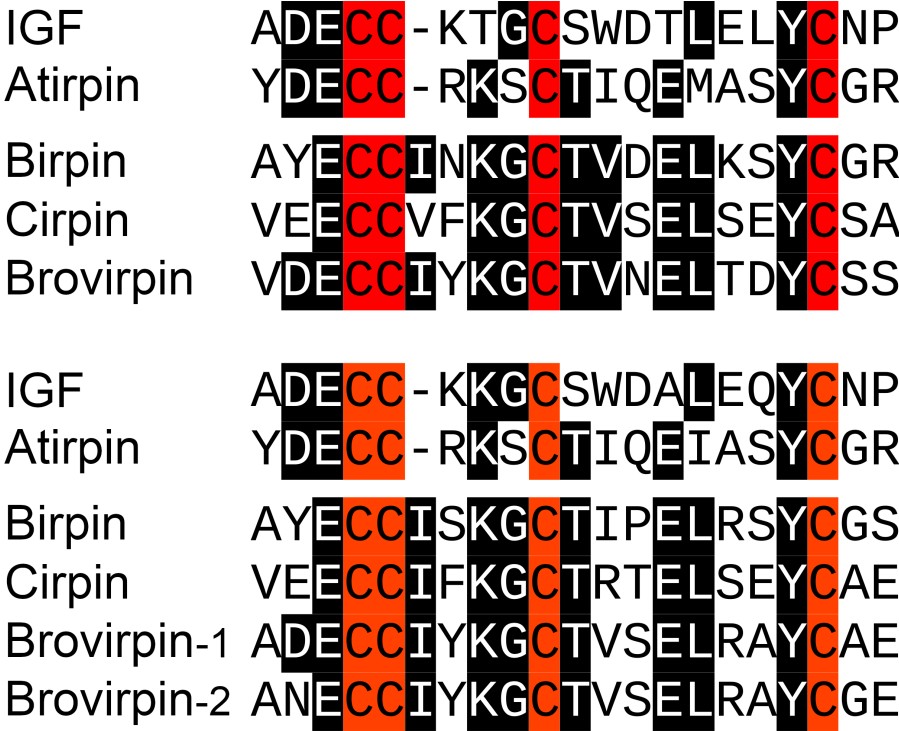

**Figure 4  IGF sirp sequence comparison.** The core sequences of the A-chain of IGF and the various sirps in *Zootermopsis nevadensis* (top panel) and *Macrotermes natalensis* (bottom panel) are aligned. Note that in both species there are three amino acid residues between the second and third cysteine residues in IGF and atirpin, while in the other sirps these cysteine residues are separated by four amino acid residues.

of the latter species show lower expression levels. This suggests that in these species the major tissue that expresses gonadulin is likely the female reproductive system, as in other insect species (*Liao & Nässel, 2020*; *Veenstra, 2020*; *Veenstra et al., 2021*; *Leyria et al., 2022*). However, as mentioned above, in several termites the gonadulin signaling pathway has been lost. The very large majority of *C. secundus* transcriptome SRAs were made from head and thorax and hence do not include the ovary and this likely explains why gonadulin expression is only very rarely observed in such SRAs (Spreadsheet S2, *C. secundus*). Alternatively, it is possible that gonadulin is not as important in this species as in the *Zootermopsis* and *Hodotermopsis*. The presence of an additional cysteine residue in the predicted gonadulin amino acid sequence of *C. secundus* is somewhat surprising as an odd number of cysteine residues can lead to poor protein folding in the endoplasmatic reticulum and might indicate loss of relevance for this hormone (Fig. S1). Nevertheless there is a significant number of gonadulin reads in three king transcriptome SRAs (ERR2615947, ERR2615948 and SRR5457747) from *C. secundus*.

*Drosophila* dilp8, a gonadulin ortholog, and dilp7 are known to act through leucine-rich G-protein coupled receptors, LGR3 and LGR4 respectively (*Colombani et al., 2015*; *Garelli et al., 2015*; *Vallejo et al., 2015*; *Jaszczak et al., 2016*; *Imambocus et al., 2022*), and it is possible that IGF might act as a ligand for LGR5, that is commonly present in

hemimetabolous insects (*Veenstra, 2020*). LGR4 and LGR5 are present in all termite species studied, while LGR3 could not be identified in the transcriptomes of *Coptotermes formosanus*, nor in those from the various *Macrotermes* and *Reticulitermes* species. The genome SRAs of *M. natalensis, R. speratus* and *C. formosanus* were searched for reads that could code for part of an LGR3. With one interesting exception, these searches either yielded no results or RNAseq reads coding for part of the LGR5 receptor. The exception concerns *M. natalensis* where reads corresponding to a piece of the original LGR3 was found. However, this sequence has undergone several mutations, including a stop coding inside the original coding sequence and hence can no longer code for a functional LGR3 (Fig. S9). This confirms that these species have indeed lost the LGR3 gene. The gonadulin gene in the same species has either been lost or mutated into a pseudogene. This concomitant loss of LGR3 and gonadulin is consistent with the hypothesis that LGR3 is the gonadulin receptor.

In *Drosophila* the gonadulin dilp8 is expressed in the developing imaginal discs and as long as it is secreted metamorphosis can not proceed (*Colombani, Andersen & Léopold, 2012*; *Garelli et al., 2012*). Gonadulin is also expressed in the female reproductive system of *Drosophila, Rhodnius prolixus* and *Locusta migratoria* (*Liao & Nässel, 2020*; *Leyria et al., 2022*; *Veenstra et al., 2021*). In the migratory locust the effects of gonadulin RNAi mimic those of starvation and it has been suggested that it acts through the central nervous system (CNS), where its receptor is expressed, to increase feeding in order to insure sufficient metabolites for vitellogenesis (*Veenstra et al., 2021*). It is interesting to note that in higher termites, where feeding by the queen no longer depends on her actively searching for food but on the number of the workers that nurse her, such a function would be redundant. This may have facilitated loss of the gonadulin signaling system in these species.

The number of reads for the dilp7 ortholog and its putative receptor LGR4 is low in all termite SRAs analyzed and hence the data analyzed here do not provide useful details with regard to its expression.

## Expression of IGF and sirps and functionl relevance of atirpin

The expression of IGF in the various termite SRAs is quite variable (Spreadsheet S2). It tends to be higher in both male and female as well as in both primary and secondary reproductives. In insects IGF is mainly expressed by the fat body, but has also been reported to be present in ant folliculocytes (*Yan et al., 2022*). In *Z. nevadensis* its expression is highest in reproducing females where the number of IGF reads in transcriptome SRAs from whole bodies reach 5 per million spots. In *R. speratus, C. secundus* or *M. natalensis* the expression of IGF is also highest in reproducing females (Spreadsheet S2). In the latter species IGF reads in queen transcriptome SRAs can be as high as 40 per million spots. In larval whole body transcriptome SRAs from *M. barneyi* larvae IGF spots are variable and range from 0.2 to 3.1 per million spots.

Atirpin is found in virtually every SRA analyzed, those from the head, head and thorax, and abdomens from all species and all castes. This sirp was previously identified in the fat body in *M. natalensis* and was called ilp9 by the authors (*Séité et al., 2022*). It is also found in the rare tissue specific termite SRAs, such as those from the salivary gland, foregut,

midgut and hindgut from *C. formosanus* and the ovaries from *Prorhinotermes simplex*. In the German as well as the American cockroach atirpin is similarly expressed in virtually all tissues (*Castro-Arnau et al., 2019*; *Veenstra, 2023*). Hence, it is likely that also in termites atirpin is ubiquitously expressed. Whereas it makes physiological sense for the fat body to secrete IGF in order to signal that growth is possible when there are sufficient energy and protein stores, it does not make much sense that every single tissue would be able to signal growth by secreting atirpin. Furthermore, if atirpin were to stimulate growth it would not explain why expression of this hormone is increased so strongly during starvation in the ovary of *Blattella germanica* (*Castro-Arnau et al., 2019*). It seems likely then, that atirpin is not a general growth hormone. Rather, it looks like atirpin might be an autocrine signal that is secreted when tissues need metabolic resources. By secreting atirpin a tissue would activate its own insulin/IGF RTK and stimulate uptake of metabolites such as glucose and amino acids. This is somewhat reminiscent of, but also different from, the function of dilp3 in the *Drosophila* midgut, where dilp3 is produced and released as an autocrine growth factor (*O'Brien et al., 2011*). Such a hypothesis may also clarify why, when *Blattella* females are starved, the expression of atirpin is increased to a much higher degree in the ovary than in other tissues (*Castro-Arnau et al., 2019*), since the ovary is metabolically the most active tissue. If atirpin is indeed an autocrine signal to insure sufficient metabolites it would make sense to act preferentially on the same receptor as IGF. This might explain why of all the sirps it is the structure of atirpin that is the most similar to IGF.

Of the other three sirps, brovirpin is perhaps the most interesting as in higher termites there are several genes coding such a peptide. Less evolved species appear to have only one brovirpin gene, as shown by the genomes of *Zootermopsis* and *Cryptotermes* and suggested by the transcriptomes of *Mastotermes* and *Hodotermopsis* (Spreadsheet S1). The reads for birpin, cirpin and brovirpin in SRAs from heads or heads and thoraces and the presence of characteristic endocrine convertase cleavages sites in their precursors suggests that these sirps are likely expressed by brain neuroendocrine cells, as is the case for such peptides in other insect species, including the American and German cockroaches (*Castro-Arnau et al., 2019*; *Veenstra, 2023*). However, unlike birpin and cirpin, brovirpin is also expressed in the abdomen. There is good evidence that the expression of brovirpin in the abdomen is limited to the ovary. Thus, the ovary transcriptomes of *Prorhinotermes simplex* mature females reveal very abundant expression of brovirpin (Spreadsheet S2, *P. simplex*), as do the SRAs from degutted primary females from *Cryptotermes secundus* that simultaneously contain numerous RNAseq reads for the vitellogenin receptor. Transcriptome SRAs from the same species that lack reads for the vitellogenin receptor also lack brovirpin reads (Fig. S10). There are no ovary specific SRAs for *Zootermopsis*, but brovirpin is found in very large numbers in reproducing females but not in males (Spreadsheet S2, *Z. nevadensis*-1). In *R. speratus* brovirpin reads are found in large numbers of whole body SRAs prepared from queens, secondary and alate females, but almost absent from alate males, kings, workers or soldiers (Spreadsheet S2, R-speratus-1). There is very little data for brovirpin expression in *Macrotermes* species and, unfortunately, there is not a single ovary specific transcriptome SRA from any *Macrotermes* species. One might expect the three degutted *M. natalensis* queen SRAs (SRR5457281, SRR5457282 and SRR5457283) to reveal significant

numbers of brovirpin reads, but they do not. However, they neither show significant numbers of vitellogenin receptor reads, so these samples may have contained very little ovarian tissue. Nevertheless, as was the case for *C. secundus*, when the number of brovirpin reads are plotted against the number of vitellogenin receptor reads in this species, large numbers of brovirpin reads are only found in SRAs that also contain large numbers for the vitellogenin receptor (Fig. S11). Thus the data strongly suggest that termite brovirpin is specifically expressed by the ovary of reproducing females. As there is no ovary tissue in the head the expression of brovirpin in the head is likely by neuroendocrine cells in the brain. Such a dual expression of a sirp by both the ovary and brain neuroendocrine cells is also observed in the case of *Periplaneta* sirp 3 (*Veenstra, 2023*). Brovirpin is also expressed in much larger quantities in female than in male soldiers in *Z. nevadensis* (Spreadsheet S2, *Z. nevadensis*-1), but brovirpin is not expressed in female soldiers from *R. speratus* (Spreadsheet S2, *R. speratus*-1).

Birpin is also well conserved in cockroaches, except that basal cockroaches usually have two birpin-like genes, while in termites there is only one. The expression of both birpin and cirpin appears to be limited to the head and both genes are thus likely expressed by neuroendocrine cells in the brain.

## Expression and functional relevance of ILP receptors

Ilps typically act on insulin/IGF RTKs. I have previously suggested that the orphan GPCR LGR5 might be an additional receptor for IGF in hemimetabolous insect species and some deuterostomes (*Veenstra, 2020*; *Veenstra, 2021*). LGR5 is present in all termite genomes and is expressed, but the data do not yield clues as the function of LGR5 or the validity of the hypothesis that LGR5 is a second IGF receptor. Nevertheless, it is interesting to note that it appears to be well expressed in the ovary specific SRA from *P. simplex* (Spreadsheet S2, *P. simplex*).

Cockroaches have been reported to have three insulin tyrosine kinase receptors (*Kremer, Korb & Bornberg-Bauer, 2018*; *Smýkal et al., 2020*). Although *Periplaneta americana* possesses four such receptors (*Veenstra, 2020*), termite genomes and transcriptomes have no ortholog of this additional receptor as shown by a phylogenetic tree revealing the fourth *Periplaneta* receptor to be unique and without a termite ortholog (Fig. S12).

The existence of three insulin/IGF RTKs suggests that their functions are different. Assuming for simplicity that one would be for body growth and a second one for reproduction one would still expect expression of both types of receptors in a plurality of tissues, the ovary *e.g.*, will need to grow before it can produce eggs. So it is not surprising that the few existing tissue specific termite transcriptome SRAs show expression of more than one receptor, as was also reported for ants (*e.g.*, *Yan et al., 2022*). Others have reported that in *Prorhinotermes simplex* expression of the three insulin/IGF RTKs varies between the different body parts (*Sang et al., 2016*; *Smýkal et al., 2020*) and this is generally the cases in termites.

In the beetle *Gnatocerus cornutus* there is good evidence that IGF acts preferentially through the insulin/IGF RTKA (*Okada et al., 2019*), the ortholog of the functional insulin/IGF RTK of *Drosophila*. RTKA (cluster I from *Smýkal et al., 2020*) is therefore

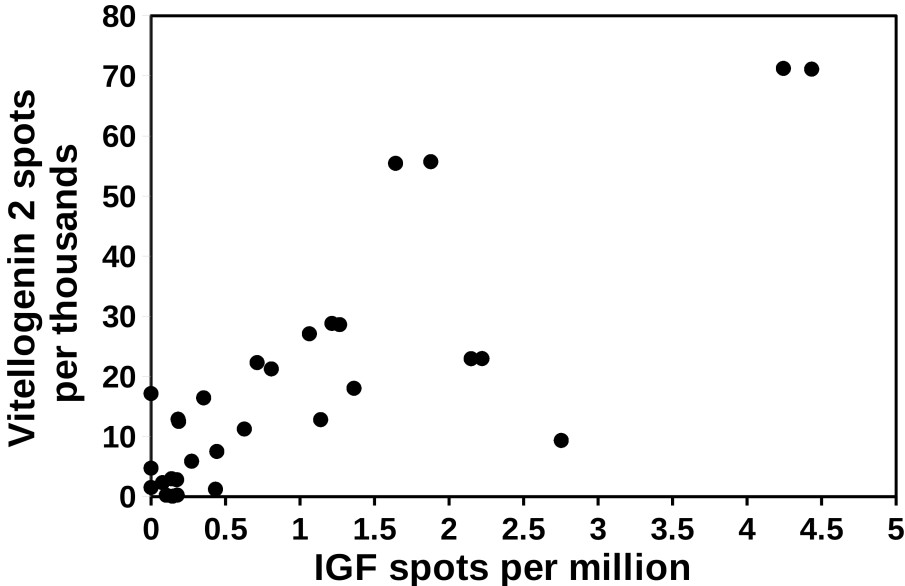

**Figure 5  Correlation between the expression of vitellogenin and IGF in *Cryptotermes secundus*.** The relative number of IGF spots were plotted against the number of vitellogenin 2 spots in the same SRA. Note that there appears to be a positive correlation between the two. SRAs used for this figure are listed in the Supplementary Data.

likely the common insect IGF receptor. If atirpin is indeed an autocrine signal to insure sufficient metabolites it would make sense to act preferentially on the same receptor as IGF. As discussed above the structural similarity between atirpin and IGF in the core ilp sequence also suggests that atirpin acts on the same receptor as IGF. If this hypothesis were correct, it would suggest that the other three sirps act preferentially on the insulin receptors RTKB1 and RTKB2.

### IGF and Sirps in relation to growth, vitellogenesis and salivary gland protein production

In insects both JH and ilps are necessary for vitellogenesis (*Roy et al., 2018*). JH is produced by the corpus allatum and the size of this gland increases dramatically in *Macrotermes* queens as they become physogastric (*Lüscher, 1976*). Since the effects of IGF on vitellogenesis are unknown, it seemed of interest to see how the synthesis of vitellogenin might correlate with that of IGF. Sufficient data to explore this conjecture are available for two species, *C. secundus* and *M. natalensis*.

A preliminary examination of the *Cryptotermes* data shows a good correlation between the expression of IGF and vitellogenin 2 in this species (Fig. 5). Both IGF and vitellogenin are produced by the fat body (*Hagedorn & Kunkel, 1979*; *Veenstra, 2020*) and hence, a causal relation between the two seems unlikely from a physiological point of view. Indeed, it is more likely that both react to the same external conditions. As described below relative spot numbers for both IGF and vitellogenin decrease as atirpin expression increases.

A correlation between the expression of vitellogenin and IGF for *Macrotermes* yields different results. The *Macrotermes* data come from a study where transcriptome data were collected from queens of different ages. In this species it is the age of the queen that determines the number of vitellogenin 1 spots in fat body SRAs rather than the number of IGF spots (Fig. 6). A similar graph for vitellogenin 2 and IGF yields qualitatively similar results (Fig. S13). The drawings of the queens of the five different age groups suggest that the sizes of the Q0, Q1 and Q2 queens are virtually identical while the size of the Q3 and Q4 queens are much larger. This is confirmed by the size and weight data reported for these termites (Supplementary Data from *Séité et al., 2022*). The data imply that the relative growth of the abdomen of the queen is highest between the Q2 and Q3 stages. Whereas Q1 colonies contain on average seven workers, their numbers have increased to around 57 in Q2 colonies (Supplementary Data from *Séité et al., 2022*). Thus at the Q2 stage the contribution of food provided to the queen by the workers has become significant and this likely explains both the increase in vitellogenin mRNA and the start of physogastry. As IGF spots are highest in Q2 queens it is possible that IGF, as in other insect species (*Slaidina et al., 2009*; *Okamoto et al., 2009*; *Defferrari, Orchard & Lange, 2016*), stimulates body growth.

In *Cryptotermes* there is a negative correlation between the spots for atirpin and those for vitellogenin 2 (Fig. 7). The latter is consistent with the hypothesis that atirpin is an autocrine that is secreted when a tissue lacks sufficient metabolites rather than it being a growth hormone. This hypothesis is also congruent with the observation that large numbers of atirpin in a transcriptome are only found when the number of IGF spots is below a threshold of about 0.2 reads per million (Fig. 8). Adipokinetic hormone (AKH) is a neuropeptide that mobilizes energy substrates and is typically released during flight and starvation. Reproducing queens have lost their wings and *a priori* are not expected to be starved. Nevertheless, during the time it may take to collect a queen and dissect its fat body she may not be nursed very well and perhaps not at all; such queens may well start to show signs of starvation. Plotting the number of AKH spots against those of vitellogenin 2 in *Cryptotermes* similarly shows a negative correlation (Fig. 9). Unlike IGF and atirpin, AKH is secreted by the regulated pathway and is stored in significant quantities in the corpora cardiaca. Although the initial response of starvation is likely the secretion of stored hormone, the starvation response appears to be sufficiently strong to increase transcription of the AKH gene. If both AKH and atirpin are released under stress one should expect a positive correlation between expression of these two genes and this is what is found (Fig. S14).

In insects both ilps and juvenile hormone are needed for vitellogenesis. One should expect this also to be the case in termites. In physogastric queens the relative volume of the brain compared to that of the abdomen becomes very small, so small in fact that brain hormones need to be produced in much larger quantities than in a cockroach with the same brain size. It is therefore not surprising that the size of the corpus allatum in physogastric queens increases enormously in order to produce sufficient juvenile hormone (*Lüscher, 1976*). Unlike the corpus allatum, an isolated gland that has the space to increase its size, the protocerebral neuroendocrine cells are constrained in space and can not easily increase

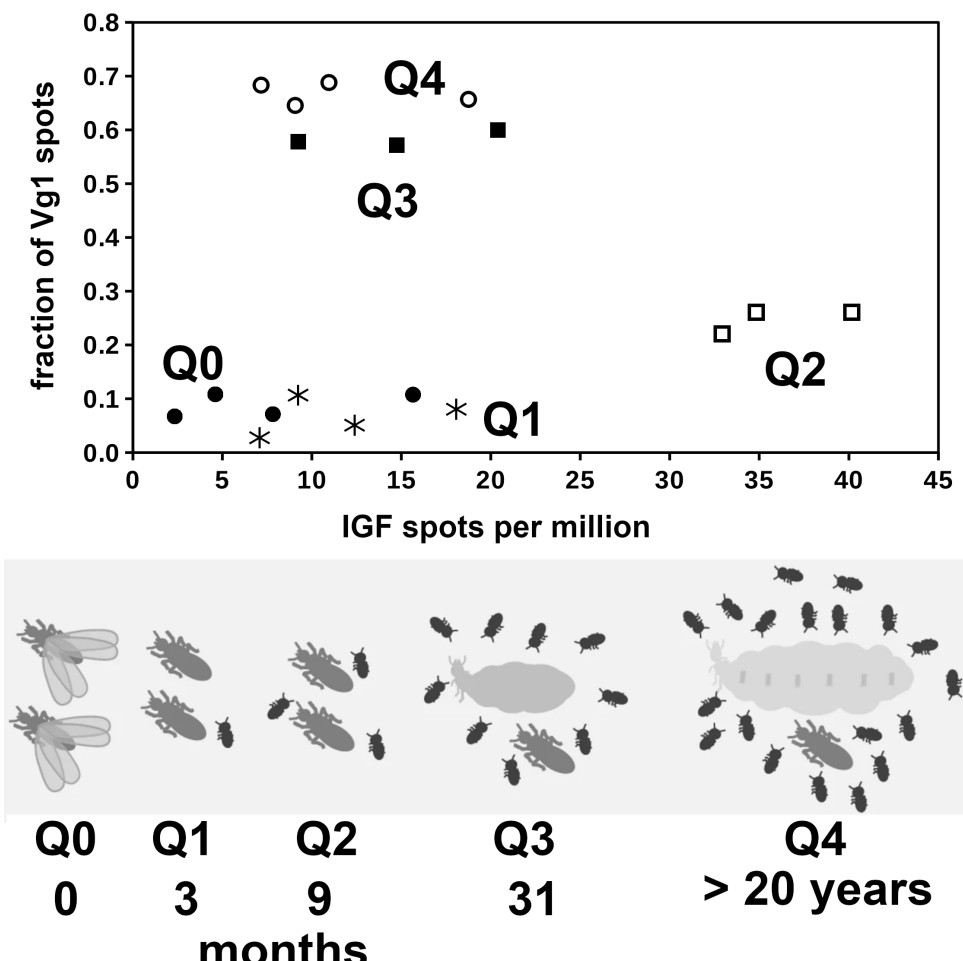

**Figure 6** **Correlation between the expression of vitellogenin and IGF in *Macrotermes natalensis*.** Top panel shows the correlation between the number of IGF and vitellogenin 1 spots in *M. natalensis* queens of different ages, Q0 (closed circles), Q1 (asterisks), Q2 (open squares), Q3 (closed squares) and Q4 (open circles). The age and the approximate size of the queens are indicated in the bottom panel and the increase in the size of their colonies is symbolically represented by the number of larvae drawn. Note that in Q0 and Q1 queens the fraction of vitellogenin 1 spots is still low, but that it increases significantly in Q2 queens but only becomes very high in Q3 and Q4 queens. Also note that IGF expression appears highest in Q2 queens. The bottom panel of this figure was copied from figure 1 from *Séité et al., 2022*.

their size. It is thus likely that brovirpin produced by the ovary is the sirp that stimulates vitellogenesis. This also implies that it is unlikely that birpin or cirpin are important for vitellogenesis in physogastric queens. As discussed below, birpin may well play an important role in caste determination.

Workers do not reproduce and so there is no need for vitellogenesis, instead workers need to feed small larvae, nymphs, royals and soldiers. Both stomodaeal and proctodaeal trophollaxis are well documented in termites. In stomodaeal trophollaxis both partially digested food and salivary enzymes are shared, in higher termites this consists exclusively of salivary gland secretions (*Grassé, 1982*). Given the scarcity of protein in wood, the

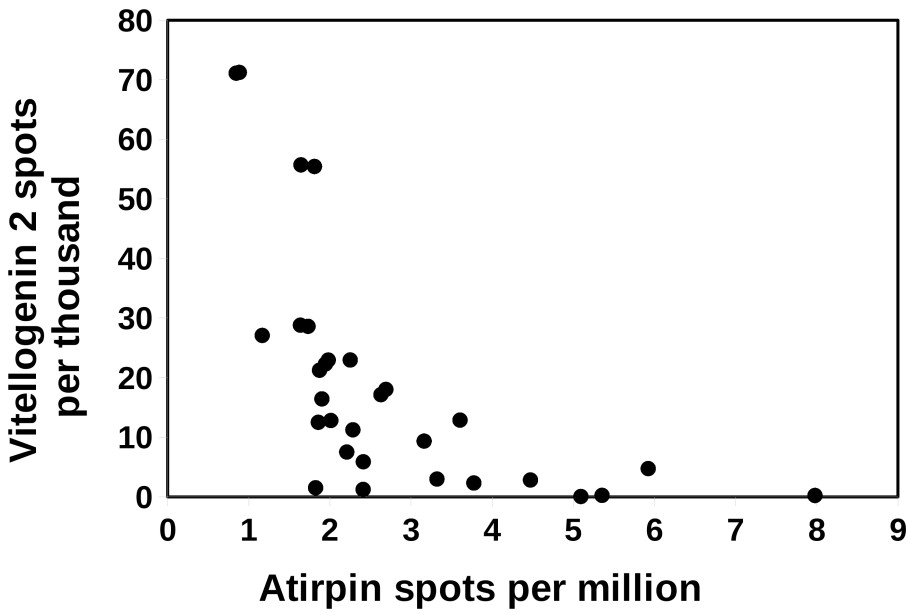

**Figure 7** **Correlation between vitellogenin and atirpin in *Cryptotermes secundus*.** The number of atirpin spots were plotted against the number of vitellogenin 2 spots in a number of SRAs from *C. secundus*. Note that an increase in atirpin expression is associated with a decrease in vitellogenin 2 expression. The data is derived from the same SRAs as in Fig. 5.

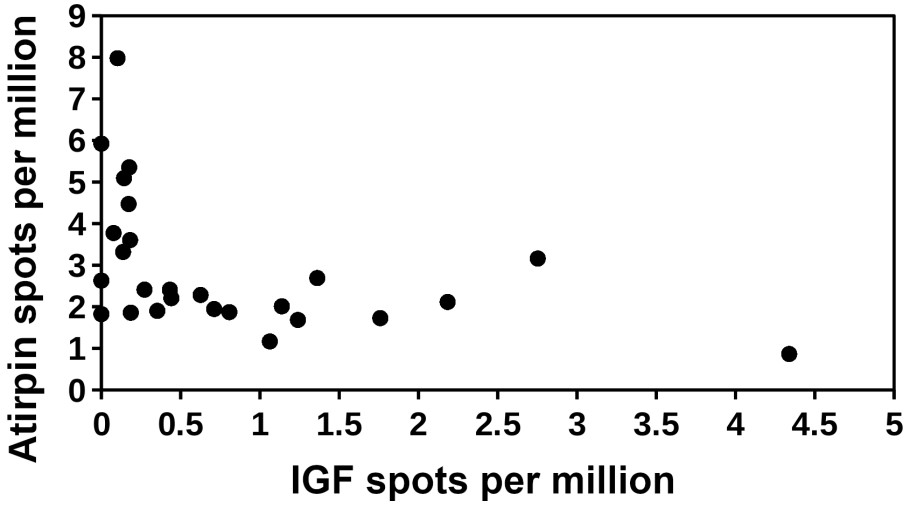

**Figure 8** **Correlation between IGF and atirpin in *Cryptotermes secundus*.** The number of atirpin spots were plotted against the number of IGF spots in a number of SRAs from *C. secundus*. Note that high numbers of atirpin spots are only observed when the number of IGF spots is low. The data is derived from the same SRAs as in Fig. 5.

enzymes from the salivary gland that are fed to other members of the colony constitute an important protein source. In workers protein production by the salivary glands must be very significant as a fraction of total protein synthesis and it is a reasonable hypothesis that

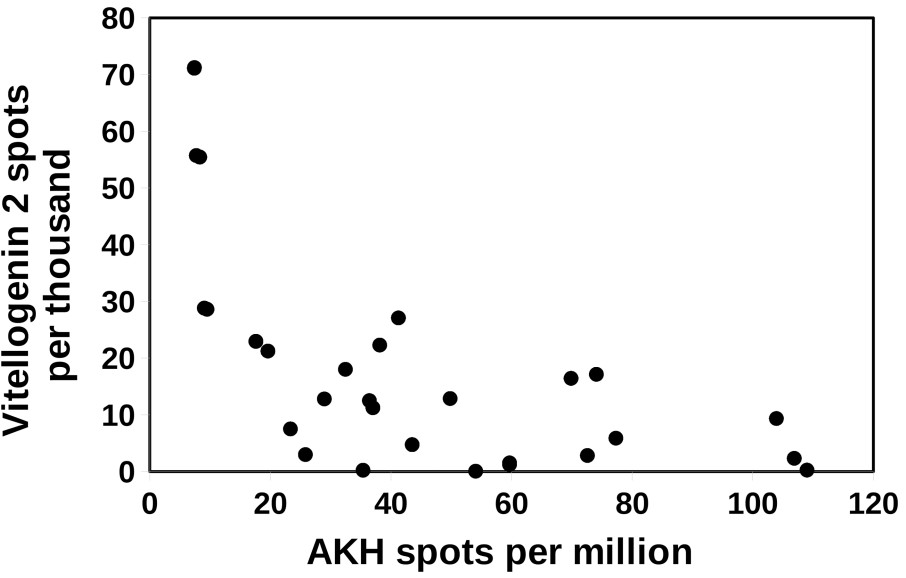

**Figure 9 Correlation between AKH and vitellogenin 2 in *Cryptotermes secundus*.** The number of vitellogenin 2 spots were plotted against the number of AKH spots in a number of SRAs from *C. secundus*. Note that highest numbers of vitellogenin spots are observed when the number of AKH spots is low. The data is derived from the same SRAs as in Fig. 5.

this increased protein production by the salivary glands in workers might be stimulated by one or more ilps. In soldiers both mandibles and salivary glands are transformed into defensive apparatus and consequently this caste needs to be fed by the workers as well and the salivary glands in soldiers have no need to continuously synthesize proteins. There is relatively little data on the expression of ilps in the soldiers. Nevertheless, it is noteworthy that in the heads of soldiers from either *R. aculabialis* or *R. speratus* there is no significant expression of any sirp, while in worker heads cirpin is the most abundantly expressed sirp (Spreadsheet S2). In *Z. nevadensis* and *M. darwiniensis* where the numbers are much smaller, the soldiers similarly neither express cirpin, while workers do. So it seems possible that cirpin is responsible for the stimulation of protein synthesis in the salivary gland.

## Expression of neuropeptides

The relatively large number of transcriptome SRAs for *R. speratus,* 15 different SRAs for 6 month old queens and 15 SRAs for 6 month old kings, offers the possibility to test the differences in neuropeptide expression between male and female reproductives. Fifty nine neuropeptides were tested using Wilcoxon rank test (Spreadsheet S2, *R. speratus-*2). When performing large numbers of statistical tests it is to be expected that some will by pure chance yield p's that may look significant. Thus, a hundred random tests is expected to yield on average 5 tests with a $p < 0.05$. Results for eclosion hormone 2 and glycoprotein B5 resulted in $p$ values of .02088 and 0.02382 and were therefore ignored. Values for other neuropeptides are smaller, sometimes very much so. These are allatostatin A ($p < 0.00424$), allatotropin ($p < 0.00374$), CNMamide ($p < 0.00374$), hansolin ($p < 0.00001$), neuropeptide F1 ($p < 0.00012$), atirpin ($p < 0.00142$) and brovirpin

($p < 0.00001$). The differences between the expression of hansolin and bovirpin are particularly impressive. In the case of brovirpin this probably reflects its expression in the ovary. It is likely that hansolin is similarly expressed in the ovary, as indicated by the *P. simplex* ovary SRA, but like brovirpin, it is also expressed in the brain. Allatostatin A and neuropeptide F1 are more highly expressed in kings than in queens, while expression of the other five is higher in queens than in kings. The differential expression of these seven neuropeptides between 6 month old kings and queens may represent characteristic differences between reproductive males and females of this species. Several of these peptides show similar differences in expression when comparing alate males with alate females or primary kings with secondary queens.

## Caste determination

There is a lot of interest with regard to the regulation of caste differentiation in termites. In *Z. nevadensis* birth order is of the utmost importance. The first born larva of a royal *Zootermopsis* pair is well fed and as a third instar molts into a presoldier and subsequently into a soldier, while its younger sibling receives less food and is destined to become a worker. In this species neural lazarillo, a lipocalin, was identified as a crucial factor that might determine whether or not a third instar larva would develop into a worker or a presoldier (*Yaguchi et al., 2018*). In none of the species a correlation between the expression of neural lazarillo and caste determination was found. As this lipocalin is expressed in the gut, it is perhaps its diminshed expression induced by RNAi that leads to a decrease in food uptake and thereby interferes in the progression of a larva into a presoldier.

Hormones, rather than a structural protein, may be better candidates for transmitting pheromone signals into regulators of polyphenism. Indeed, the role of JH in the induction of soldiers is well documented for several species (*e.g.*, *Lüscher, 1958*; *Lüscher, 1972*; *Lelis & Everaerts, 1993*; *Masuoka et al., 2015*). The role of the ilps in this process is suspected (*e.g.*, *Hattori et al., 2013*; *Kremer, Korb & Bornberg-Bauer, 2018*), but needs supporting data, some of which can perhaps be extracted from published transcriptomes. The most extensive data set on caste determination comes from the previously mentioned work on *Zootermopsis* (*Masuoka et al., 2018*; *Yaguchi et al., 2018*; *Yaguchi et al., 2019*).

There are SRA data from three experiments (*Masuoka et al., 2018*; *Yaguchi et al., 2018*; *Yaguchi et al., 2019*) in which gene expression was determined at various time points before and after the third molt of these two siblings. The latest of these three experiments contains the richest data set. The data suggests a higher expression of birpin, ETH and PDF in the presoldiers, but variability is very high (Spreadsheet S2, Z.nevadensis-2). Whereas the differences in apparent expression of birpin seems to be consistent at the different time points, those for ETH and PDF are only found later in the cycle. The time to the next molt is much shorter in the presoldier than in the worker destined larva, probably as a direct result of it being better fed. Therefore, the differences in expression in ETH and PDF might be more apparent than real and reflect the faster development of the presoldier larvae. On the other hand in the fruitfly ETH is known to stimulate juvenile hormone production (*Meiselman et al., 2017*). If this neuropeptide would similarly stimulate juvenile hormone
production in termites, then ETH might be directly responsible for the increase in juvenile hormone and the induction of the soldier phenotype.

A second data set come from *Reticulitermes labralis*. In the absence of a queen, workers of this species can develop into neotenic reproductives. Four weeks after isolating 50 late instar larvae together with two soldiers, a neotenic reproductive appears (*Ye et al., 2019*). Transcriptomes were obtained from the heads of workers from a colony with a queen, workers isolated for 1, 2 or 3 weeks and neotenic reproductives four weeks after starting the experiment. In the isolated workers, expression of the two storage proteins increases rapidly, but changes in the expression of ilps and neuropeptides are too small to be considered significant. In the new neotenic reproductives the expression of hansolin, brovirpin and vitellogenin receptor are all strongly increased, while that of cirpin has decreased. Interestingly, the relative expression of the RTK-B2 receptor has also increased, suggesting that is important for the regulation of reproduction and/or vitellogenesis (Spreadsheet S2, *R. labralis*).

A single study on caste determination in *Macrotermes barneyi* (*Sun et al., 2019*) is perhaps the most informative. *M. barneyi* is a higher termite, where caste determination is very much foreordained. Whole animal transcriptomes were produced for five immature castes predestined to develop into reproductive adults, major or minor soldiers, or major or minor workers. The expression of storage proteins and vitellogenin of these five castes of *Macrotermes barneyi* (*Sun et al., 2019*) shows that the nymphs distinguish themselves. They have the highest number of spots for met-hexamerin, consistent with the more thorough tissue remodeling. Expression is about ten times higher in nymphs than in the other castes. The expression of tyr-hexamerin is also highest in nymphs, but their relative expression is "only" about three times as high as in the presoldier castes and more than five times as high as in the preworker castes. This is consistent with the nymphs being better nourished (Spreadsheet S2. *M. barneyi*-2).

The number of IGF spots in these SRAs are quite variable both between and within the five castes and are therefore hard to interpret. Of all the ilps there are, the differences in birpin expression that are the largest and most consistent between the five different castes (Fig. 10). Both major presoldier and nymph SRAs have large relative numbers of birpin spots, with the nymph SRAs having by far the most. Nymph SRAs have about 17 times higher relative birpin numbers than the SRAs from minor presoldiers or workers, and still almost three times as many as the major presoldier SRAs. In both nymphs and major presoldiers atirpin is significantly increased. These two castes differ in the expression of allatostatin A. The relative number of allatostatin A spots in nymphs is about five times larger than in the other four castes. Allatostatin A neuropeptides were identified from the cockroach *Diploptera punctata* as inhibitors of JH synthesis (*Woodhead et al., 1989*; *Pratt et al., 1989*). The expression of allatostatin A in the brain of females is increased about three-fold as synthesis of JH is inhibited at the end of the vitellogenic cycle (*Garside, Bendena & Tobe, 2003*). As JH induces the soldier phenotype, inhibition of JH synthesis in nymphs is thus to be expected. The SRAs used here contain not only the brain but the whole insect, and allatostatin A is also abundantly expressed by the ventral nerve cord and

| Gene | MinPreWor | MajPreWor | MinPreSol | MajPreSol | Nymphs | Differences | |
|---|---|---|---|---|---|---|---|
| Hansolin | 0.0015 | 0.0014 | 0.0012 | 0.0017 | 0.0074 | Nymphs | 5.09 |
| AstA | 0.0226 | 0.0216 | 0.0201 | 0.0185 | 0.0983 | | 4.75 |
| Sulfakinin | 0.0015 | 0.0011 | 0.0015 | 0.0017 | 0.0007 | | 1.98 |
| | | | | | | | |
| Birpin | 0.0011 | 0.0009 | 0.0007 | 0.0058 | 0.0162 | | 17.55 (2.77) |
| CNMamide | 0.0220 | 0.0130 | 0.0242 | 0.0805 | 0.0777 | | 4.01 |
| NPF 1 | 0.0311 | 0.0318 | 0.0309 | 0.0140 | 0.0148 | Nymphs | 2.17 |
| NPLP1 | 0.0466 | 0.0469 | 0.0454 | 0.0225 | 0.0278 | & | 1.84 |
| Fliktin | 0.1857 | 0.1783 | 0.1728 | 0.0970 | 0.1153 | MajPreSol | 1.69 |
| sNPF | 0.0182 | 0.0167 | 0.0169 | 0.0111 | 0.0098 | | 1.65 |
| AstCCC | 0.0199 | 0.0184 | 0.0202 | 0.0134 | 0.0130 | | 1.48 |
| Baratin | 0.2801 | 0.2660 | 0.2755 | 0.1870 | 0.2008 | | 1.41 |
| | | | | | | | |
| AKH | 0.0165 | 0.0177 | 0.0178 | 0.0274 | 0.0137 | MajPreSol | 1.67 |
| Myosuppressin | 0.0125 | 0.0231 | 0.0117 | 0.0075 | 0.0120 | | 3.10 (1.92) |
| | | | | | | | |
| Brovirpin all | 0.0093 | 0.0114 | 0.0033 | 0.0099 | 0.0120 | MinPreSol | 3.17 |
| | | | | | | | |
| Cirpin | 0.0006 | 0.0019 | 0.0002 | 0.0017 | 0.0010 | ???? | 4.82 (1.87) |
| | | | | | | | |
| Tyr-Hexamerin | 103,583 | 103,474 | 199,336 | 178,001 | 545,405 | | |
| Met-Hexamerin | 28,077 | 37,184 | 32,546 | 25,348 | 290,218 | | |
| Vitellogenin | 105 | 69 | 160 | 101 | 1,653 | | |

**Figure 10 Differential gene expression and caste determination in *Macrotermes barneyi*.** Differential gene expression for a number of neuropeptides and three proteins. The numbers are the averages from three separate experiments. The neuropeptide numbers are relative to the total neuropeptide spots in the respective SRAs, the numbers for the proteins are the actual number of spots in the SRAs. Arbitrarily similar values have been given the same highlight color, where orange is the highest, white the lowest and yellow intermediate. The last columns indicate which groups are different from the others, the numbers are the fold differences between the average values of the highest and the lowest groups. A second number between parentheses indicates the fold difference between the highest and the intermediate groups. Note that the apparent expression of allatostatin A (AstA), hansolin, birpin and CNMamide is much higher in nymphs.

enteroendocrine cells. A five-fold difference in the whole animal is therefore likely to be sufficient to inhibit JH synthesis in termite nymphs as well.

There are several other neuropeptide genes which show notable differences in the number of spots between the castes. In two additional neuropeptides these differences are as, or almost as large as in the case of allatostatin A. The number of hansolin spots are about five times higher in nymphs than in any of the other castes (Fig. 10). Hansolin was recently discovered as a putative neuropeptide in the stick insect *Carausius morosus* by mass spectrometry (*Liessem et al., 2018*). Its function is unknown, however the above mentioned strong expression in the ovary of *P. simplex* suggests a role in reproduction, and perhaps it is also important in caste determination. Spots for CNMamide are about four times more abundant in nymphs and major presoldiers than in the other three groups. CNMamide is a neuropeptide that was identified from central nervous system of *Drosophila* (*Jung et al., 2014*) and may be generally present in arthropods (*e.g., Veenstra, 2016*). It is also expressed by entereroendocrine cells and has recently been shown to stimulate feeding when essential amino acids are needed (*Kim et al., 2021*). As the relative number of spots are increased in both major presoldiers and nymphs, it might stimulate feeding and thereby indirectly play

a role in stimulating birpin production and release or otherwise stimulate growth in those larvae.

Even in the absence of rigorous statistical confirmation the size of the observed differences in transcript spots for birpin, allatostatin A, hansolin and CMNamide are sufficient to suggest that they are physiologically relevant and, in the case of allatostatin A and birpin, they are not only are eye-poppingly large but also consistent with their presumed physiological functions. It would be surprising if these differences could not be confirmed by a more thorough analysis.

Results suggests smaller differences in expression for several other neuropeptides. Although these differences are much smaller, and they may not all be confirmed in a more vigorous analysis, some may well be physiologically relevant. A large number of genes show similar expression levels in nymphs and major presoldiers on the one hand, and minor presoldiers and the two worker castes on the other. This suggests that nymphs and major presoldiers share physiological traits that are absent in the other three castes. Nymphs and major presoldiers increase significantly in size as compared to the others and also show higher expression levels for storage proteins, thus indicating that they consume and digest more food. This could be the underlying cause of the similarity in expression of those neuropeptides. NPF1 and sNPF have been shown in other insects to have a role in feeding. The results for myosuppressin and sulfakinin, where a single caste seems to differ significantly from all the others, are intriguing but could represent artifacts due to the small sample size.

Spots for cirpin are smallest in minor presoldiers and minor preworkers, intermedidate in nymphs and highest in major presoldiers and major preworkers. Expression of cirpin is ten times higher in the major presoldiers than in the minor presoldiers and more than three times as high in major workers as in the minor preworkers. This might indicate that cirpin plays some role in major soldier determination. No consistent differences were found in the expression of atirpin, IGF, the dilp7 ortholog or the various tyrosine kinase receptors.

## CONCLUSION

### Plausible functions of termite ilps

Termites have seven different ilps. The functions of the three original ilps, gonadulin, the dilp7 ortholog and IGF, are likely to be similar to those in other insect species. The function of dilp7 is intriguing, as both ligand and receptor are well conserved in many animal species, but the data reported here do not contribute any insights as what its function might be. As in other arthropods (*Veenstra et al., 2021*; *Leyria et al., 2022*), the expression of gonadulin seems to be particularly abundant in the female reproductive system of the lower termites, but this signaling system was lost in higher termites. The expression of termite IGF suggests that it is likely a growth hormone as in other insect species.

Although the four different sirp types are more closely related to one another than either to the three original ilps, their expression suggests that their functions differ. As discussed, the expression of atirpin strongly suggests that it is an autocrine tissue factor that is secreted when there is a lack of sufficient metabolites. In this respect it somewhat
resembles gonadulin. However, there are significant differences between gonadulin and atirpin. Whereas atirpin appears to be an autocrine, gonadulin acts at least partly as an endocrine directly on the central nervous system. Also, the little we know about gonadulin suggests that it is secreted when there is a very large need for metabolites such as when tissues go through substantial growth, while atirpin might be a much more a local and temporary homeostatic factor. Birpin seems to be the signal used by the brain to stimulate growth independently of IGF. When produced in large quantities together with JH it will produce soldiers, when produced in even larger quantities and in the absence of JH, by the increased expression of allatostatin A, it likely stimulates the development of reproductives.

It is interesting that there are both similarities and differences between the production of reproductive and worker castes in termites and honey bees. In both species differential expression of two ilps may determine whether or not development of a reproductive ensues. If the honeybee larva is fed normal food, the quality does not induce sufficient amounts of IGF by the fat body for the development of reproductive organs. The brain then releases a sirp and the larvae develops into a worker. In termites—and perhaps other cockroaches as well—the poor nutritional quality of foods makes it perhaps impossible to secrete large quantities of IGF and it is probably birpin produced in the brain that steers development into a reproductive. The abundant expression of brovirpin by the ovary suggests that it stimulates vitellogenesis, while its expression in the brain neuroendocrine cells could have the same function. The function of cirpin is clearly the most speculative, its absence in the soldiers of some species and its down regulation in *R. labralis* when there are no larvae to feed, might indicate a role in the stimulation of salivary protein production.

## Evolution of insect sirps

An atirpin-like peptide likely evolved from IGF as an autocrine to protect tissues when metabolites are scarce. It is tempting to speculate that subsequently it, or a paralog, progressed into an neuroendocrine hormone to save the insect by inducing (premature) body growth and subsequent molting during nutritional stress. Whether this is how neuroendocrine sirps evolved in insects remains to be seen, but seems to be an attractive hypothesis.

Ilps have been extensively studied in fruit flies, silk worms, several other holometabolous insects as well as a few hemimetabolous species. In all these species there is a single neuroendocrine cell type in the brain that produce one or more sirps. This may well be different in termites, as in the American cockroach at least three different neuroendocrine cell types each produce their own specific sirps (*Veenstra, 2023*). If birpin, cirpin and brovirpin are similarly produced in different cell types in the brain, it would be easy to understand how pheromone signals can directly inhibit particular cell types and thus impede the expression of specific sirps.

### Funding

This work was funded by institutional funds from the CNRS. The funders had no role in study design, data collection and analysis, decision to publish, or preparation of the manuscript.

### Grant Disclosures

The following grant information was disclosed by the author:
CNRS.

### Competing Interests

The author declares that he has no competing interests.

### Author Contributions

- Jan A. Veenstra conceived and designed the experiments, performed the experiments, analyzed the data, prepared figures and tables, authored and reviewed drafts of the article, and approved the final draft.

### Data Availability

The genome assemblies are available at NCBI (term: blattodea) and Poulsen M; Hu H; Li C; Boomsma JJ; Zhang G (2014): Macrotermes natalensis genome assembly data. GigaScience Database. http://dx.doi.org/10.5524/100057.

The SRA accessions are available in the Supplemental File.

### Supplemental Information

Supplemental information for this article can be found online at http://dx.doi.org/10.7717/peerj.15259#supplemental-information.

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
