# Peer review of "Differential expression of some termite neuropeptides and insulin/IGF-related hormones and their plausible functions in growth, reproduction and caste determination"

_PeerJ, doi:10.7717/peerj.15259_

## Round 0.1 · original submission · Major Revisions

The reviewers concur that this is an interesting paper, but a revision, as outlined, especially, by reviewer 2 is needed in order to submit a revised version of the manuscript.

Reviewer 1 ·

Basic reporting

Overview:
This manuscript uses publicly available genome assemblies and transcriptome Short Read Archives to identify, primarily, insulin-like peptides in termites. The polyphenism observed in these organisms provides an interesting model for exploring the role of such peptides in the regulation of growth and development.

This is a very comprehensive analysis with up-to-date references. The text is clear and written in professional English. All appropriate raw data/sources have been made available, the literature references are sufficient, and an adequate introduction and background are presented to support most of the assertions made.

Minor corrections/clarifications should be made, described in "Additional comments."

Experimental design

The software, parameters and especial considerations used for this study are adequately detailed.

Since the repertoire of studies on the mechanisms of polyphenism have often focused on Hymenoptera, this work lays an interesting foundation for further expanding their understanding in termites and, in turn, of the phenomenon in general.

Validity of the findings

Findings are consistent. This work is primarily exploratory and generates an array of information and accompanying hypotheses that can now be verified in future experimental works.

Additional comments

I would recommend talking more in depth/rephrasing lines 45-50. The leap from talking about physogastry in termite queens to abdominal obesity in humans is rather harsh, they are not quite comparable. The last sentence of the paragraph could use a clarification that it refers again to termite castes.

Nomenclature. Some acronyms aren’t explained in the text when they are first used. For example, ‘dilp7’ in line 243, which appears for the first time in the abstract. Similarly, the differences between sirp and ilp are briefly explained in lines 262-264, but both terms also appear earlier. You could highlight the references to previous works where you go into detail on the considerations for the classification used.

Line 492 Here the reference should be provided. And I’m not sure I understand this analysis - the actual drawings? I don’t think it is even specified that the figures are to scale in the original source? The supplementary information in the original paper contains graphs with medians of length and weight, perhaps that might have been a better reference point, or you could ask the authors about the data. Even then, it would be a stretch to try to determine growth rate based on those limited data points. I would advise against this paragraph. Ultimately, it is insufficient evidence to support or rule out the purported role of IGF in body growth or the correlation of its expression with vitellogenin.

Line 510 It might be more accurate to describe AKH as a hormone involved in mobilizing energy to processes where it is required. Of course, that may be responding to starvation or flight, but could there be other aspects of the queen metabolism that affect AKH transcription, such as pheromones production? See: Lebreton et al 2016 doi: 10.3389/fevo.2015.00151.

Line 652 Possible functions have been described recently, see Peng, Yin, & Huang, 2021. doi:10.1038/s41392-021-00708-y.

Line 733 It is a common notion, but its lack of universality is nowadays well known, not only in termites.

Line 344 Imambocus et al., 2022 is missing in the References.

Lines 414, 437, 730 For “unpublished data” please specify the year of data collection.

Line 454 Is this reference correct, or properly placed? Kremer, Korb & Bornberg-Bauer, 2018 refer to the receptors as InR1,2 and 3.

Line 461 “…that one cluster” would improve readability.

Line 51 There’s an extra “the”

Line 61 “It is…”

Line 102 There’s an extra “analyzed”

Line 109 Please clarify with a/b to which of your previous works you are referring to.

Line 131 “Depot” instead of “store” could improve readability.

Line 192 Straightforward

Line 270 Fragment

Line 328 “Lower” might be more appropriate.

Line 401 Than

Line 483 A causal

Line 564 Remove “it is also”

Line 584 A structural

Line 628-629 “Of all the ilps there are, the differences in birpin expression are the…”

Figure 6 Remove “has”

Figure 10 Lowest

Figure S10. and S11. “Note that only those SRAs” or “that all SRAs”

Figure S13. Since Figure 6 is in another file, please write the complete description here again.

Reviewer 2 ·

Basic reporting

The introduction sounds very coloquial, it sounds imprecise in the evolutionary and social insect terminology and it is lacking citations in some of the paragraphs. There are far too many errors to list out, but here are some examples. It would be beneficial to have someone more familiar with social insects and evolution to revise the introduction.

Here are a few examples:

First Paragraph: Lines 43-45: The text is very anthropomorphized in the description of termites. Line 45: We don’t talk about one group being more evolved than other. All organisms are evolved the same amount. We talk about characters being more basal or derived or how various taxa are related to each other. e.g. sister to each other, monophyletic/paraphyletic, etc.
Lines 47-49 I don’t think termite queens are considered obese, could the author support this claim? Also, this paragraph is lacking citations.
Line 51: My understanding is that termites are not related to cockroaches. Termites are cockroaches. The entire paper should change the writing to reflect this.
Lines 64-77 is lacking appropriate citations.
Line 78, can the author provide references that support the claim that polyphenism in termites is the most sophisticated?

In the Materials and Methods section

Lines 128-130 are lacking citations.
Lines 181-182 the author mentions that samples that were labelled as fat bodies seem to contain other tissues such as the brain and the ventral cord as well. This seems like a very important problem, however, the author doesn’t explain how this problem was detected. Overall, I believe that the author should include a table that specifies the details about the original samples used for obtaining the transcriptomes (that he used for the analysis) so that we can have an idea of how clean the data is, as the author mentions possible problems with tissue labelling, termite nutritional status, etc (see lines 180-190) and the author seems to assume many possible problems with the data, but doesn’t provide enough evidence for it.
Line 191: It is not clear to me how the spots numbers are obtained. Could the author provide more details and references in this section?


Figures

Labelling and figure legends could be improved.

Figure legend 1:
The title should be more accurate. Suggestion: Simplified phylogenetic tree showing the relationships among termite genera and other cockroaches.

The phrasing of the legend is not very clear. The second sentence doesn’t make any sense.

Figure Legend 2:
The title of the figure should be more descriptive.
The legend implies that termites are distinct from cockroaches, when they are cockroaches. This should be fixed throughout the paper.

Figure Legend 3:
Phrasing could be improved for clarity. Add what the numbers mean.

Figure Legend 5: For brevity, I suggest moving the SRAs used, to the methods section or to a separate table and refer to it on the figure legend.

Figure 6
For clarity more than one color or more shapes could be added per group of different ages.
Figure 6 legend: requires some editing for clarity.
Figure 10: Labelling of all columns would improve the table.

Experimental design

1) I believe that the author should include a table with information about the samples, their quality and treatments that were used in the original studies to provide confidence in the resulting analysis.

2) Possibly due to my lack of expertise but, could the author explain better what spots are and what exactly they mean in terms of expression?

Validity of the findings

Results and Discussion

In general, the discussion about possible functional relevance of the different ILPs, their receptors, and some neuropeptides, as well as their location seems interesting. However, with no other lines of evidence together with no information about the quality and specificity of the samples made me less enthusiastic about making any claims. Maybe if the author describes the quality of the original data better this analysis will sound more reasonable.

Additional comments

Discovering the ILPs of termites as well as their possible functions are an important topic. However,
I think that the presentation is not up to the required standards of a publishable manuscript.

·

Basic reporting

In this study, Veenstra provided interesting insights into the functional differentiation of insulin superfamily peptides in termites based on amino acid sequence analysis and expression analysis. He suggested several important hypotheses suggesting that the seven ILPs in termites as gonadulin, insulin-like growth factor (IGF), relaxin, and short IGF-related peptides (sIRPs), a distinct group of insulin family genes, have different functions.

Experimental design

This is an elegant and detailed study dealing with evolutionary aspects of the highly complex and divergent insulin/IGF/relaxin system in insects.

Validity of the findings

I have no doubts about the quality and rigor of the data presented and see no need for additional analysis.

Additional comments

The author can cite Okamoto et al., Gen Comp Endocrinol, 2012, and Fujinaga et al., Sci Rep. 2019 for discussing the expression and function of insect IGF-like peptides for ovarian development.

---

## Round 0.2 · Minor Revisions

Please heed the remaining comments from both reviewers, specially including the revised figure 6 in a new resubmission of the manuscript.

Reviewer 1 ·

Basic reporting

No comment

Experimental design

No comment

Validity of the findings

No comment

Additional comments

I thank the author for his response.

Minor details:
Figure 6 was apparently modified to improve clarity (as requested by another reviewer, and denoted in the new figure legend), but Figure 6 in the file "peerj-reviewing-79864-v1" is the same as in "peerj-reviewing-79864-v0."

Other than that, the author has sufficiently addressed the issues raised, and as previously stated, this article represents an insightful contribution to the field.

Reviewer 2 ·

Basic reporting

This revised manuscript includes some important text changes and clarifications that improved the quality of the manuscript. Although I remain somewhat skeptical about the functional interpretations, I do believe that describing the expression patterns of different insulin like-peptides as well as other neuropeptides from transcriptomic data in termites is valuable and the author’s hypotheses regarding functionality would be worth testing. Therefore I support publication of this study and I have only some minor points.

-First paragraph lines 44-51 (add references)

-Line 47: please revise to “…, which in some species includes cultivating specific fungi”

-Lines 48-49: Please rephrase “As the progenitors of all members of their colony the royal pair is at the center of it all”. This sentence is a bit vague and maybe misleading. It might be better to describe specifically the role of reproductives. e.g. The royal pair is in charge of most of the reproduction of the colony.

-Line 50: Rephrase: The term “out of proportions” is not precise and is unclear. e.g. a quantity that is possible because of her specialized physiology and morphology. Termite queens have very large abdomens, a phenomenon known as physogastry, that allow them to produce thousands of eggs.

-Lines 51-52: Rephrase: This is not necessarily obvious to everyone. e.g. Why and how the same genome produces such different phenotypes is a fascinating question.

-Lines 495-496: Add reference after “Both IGF and vitellogenin are produced by the same cells”

-Figure 1: There seems to still be a problem with the figure legend.
Missing words. What is gn?

-Figure 6: The figure legend was updated, but the figure itself was not. The symbols remained the same as in the previous version, please update.

-Figure S13: Could the author please include more than two symbols here as well? This would improve clarity.

Experimental design

no comment

Validity of the findings

no comment

---

## Round 0.3 · accepted · Accept

Your paper has now been accepted for publication in PeerJ.

Reviewer 1 ·

Basic reporting

No comment.

Experimental design

No comment.

Validity of the findings

No comment.